# ON THE ROBUSTNESS OF LATENT DIFFUSION MODELS

## ABSTRACT

Latent diffusion models have achieved state-of-the-art performance on a variety of generative tasks, such as image synthesis and image editing. However, the robustness of latent diffusion models is not well studied. Previous works only focus on the adversarial attacks against the encoder or the output image under white-box settings, regardless of the denoising process. Therefore, in this paper, we aim to analyze the robustness of latent diffusion models more thoroughly. We first study the influence of the components inside latent diffusion models on their white-box robustness. We find out that the denoising process, especially the Resnet, is the most vulnerable to adversarial attacks. In addition to white-box scenarios, we evaluate the black-box robustness of latent diffusion models via transfer attacks, where we consider both prompt-transfer and model-transfer settings and possible defense mechanisms. We conclude that the adversarial examples can transfer across model structures and prompts. The adversarial vulnerability is inherited with the development of Stable Diffusion models, and the adversarial attacks are still effective when possible defenses are present. Additionally, analyzing the robustness of latent diffusion models needs a comprehensive benchmark dataset, which is missing in the literature. Therefore, to facilitate the research on the robustness of latent diffusion models, we propose two automatic dataset construction pipelines for two kinds of image editing models and release the whole dataset. Our code and dataset are available at `https://anonymous.4open.science/r/Robust-LDM-7D36`.

## 1 INTRODUCTION

Diffusion models have achieved state-of-the-art performance on a variety of generative tasks, such as image synthesis and image editing Saharia et al. (2022); Ho et al. (2020). Among various diffusion models, latent diffusion models Rombach et al. (2022) stand out for their efficiency in generating or editing high-quality images via an iterative denoising process in the latent space. Specifically, a latent noise is sampled from a uniform Gaussian distribution and denoised through the latent diffusion model step by step to form a high-quality image. Furthermore, latent diffusion models enable image editing through conditioning, which adds the embedding of a given image on the sampled latent noise to do image editing. The image editing diffusion models have been widely deployed in real-world applications such as DALL·E2 Ramesh et al. (2022) and Stable Diffusion Rombach et al. (2022). Therefore, we focus on the image editing diffusion models in this paper.

Despite the dazzling performance of latent diffusion models, recent studies show that they are susceptible to adversarial attacks Salman et al. (2023), which add human-imperceptible noise to a clean image so that latent diffusion models will incorrectly edit the input image or generate low-quality images. Therefore, for the sake of producing better diffusion models, it is important to analyze the robustness of latent diffusion models.

However, the robustness of the diffusion models has not been well studied in the literature. First of all, previous studies only focus on attacking the encoder Zhuang et al. (2023) or distorting the output image Salman et al. (2023), without taking the core structure of diffusion models (the denoising process) into consideration. Second, prior works only pay attention to the white-box setting, where attackers can obtain the full information of the victim model, like model structures and weights Goodfellow et al. (2014); Kurakin et al. (2016); Madry et al. (2017). However, they neglect the black-box setting, which assumes that attackers do not know the internals of the models and is closer

to the real-world settings Dong et al. (2018). Lastly, analyzing the robustness of latent diffusion models needs a high-quality and publicly available benchmark, which is still missing in the literature.

Therefore, we endeavor to solve the aforementioned problems in this paper. To evaluate the robustness of latent diffusion models, we deploy adversarial attacks under various white-box and black-box settings. To facilitate the research on the robustness of latent diffusion models, we propose two automatic dataset construction pipelines to generate test cases for two categories of image editing diffusion models. The contributions of our work are threefold:

- We launch adversarial attacks on representative modules inside the latent diffusion models to analyze their adversarial vulnerability. In addition, we compare the white-box robustness of two categories of latent diffusion models for image editing. We find that the denoising process, especially the Resnet, is the most vulnerable to adversarial attacks.

- We also study the black-box robustness of latent diffusion models via transfer attacks, where we consider both prompt-transfer and model-transfer settings and possible defense mechanisms. We conclude that adversarial examples transfer well across model structures and prompts. Especially, SD-v2 is more vulnerable than SD-v1, and adversarial vulnerabilities inside SD-v1 are inherited by SD-v2. Additionally, adversarial attacks are still effective against defense methods.

- We propose automatic dataset construction pipelines for analyzing the robustness of two kinds of image editing latent diffusion models. The dataset consists of 500 data pairs (image + prompt) for image variation models and 500 data triplets (image + prompt + mask) for image inpainting models. We make the code and the built dataset publicly available.

## 2 RELATED WORK

### 2.1 DIFFUSION MODELS

Diffusion models Ho et al. (2020); Rombach et al. (2022) have shown state-of-the-art performance on generating realistic images recently. They are capable of generating images or editing images via textual prompts. The core of the diffusion models is the diffusion process, which is a stochastic differential denoising process. We suppose the image $x$ is from the real image distribution, and the diffusion process gradually adds a small Gaussian noise on the image for $T$ steps. The image of the $(t+1)$-th step $x_{t+1}$ is exactly $\alpha_t x_t + \beta_t \epsilon_t$, where $\epsilon_t$ follows a Gaussian distribution, and $\alpha_t$ as well as $\beta_t$ are the parameters. By gradually adding noise on the image, the image will approximate the Gaussian distribution $\mathcal{N}(0, I)$. The denoising process is to reverse the adding noise process to generate high-quality images starting from the uniform Gaussian distribution. Therefore, diffusion models predict the noise added on the step $t$ with the input $x_{t+1}$ to remove the noise.

However, diffusion models implement the denoising process on the image level, leading to low efficiency and high computational resource costs. Then, latent diffusion models Rombach et al. (2022) are proposed to enhance the efficiency by doing the denoising operation on the latent space instead of directly on the images. Specifically, latent diffusion models utilize a variational autoencoder to transform the image between the image and latent space. Furthermore, the prompt embedding can be fused into the latent space via the cross attention mechanism in the denoising process for prompt-guided image generation. In addition to image generation, latent diffusion models enable image editing through conditioning. The denoising network initializes the latent feature with a combination of the given image embedding through the encoder and a random sampled noise, while the input prompt is the instruction on how to modify the given image. This modified version of the latent diffusion model is called the image variation model. Another category of the latent diffusion model for image editing is the image inpainting model, which is conditioning on both a given image and a mask of the keeping region. The image region outside the mask will be modified by the diffusion models.

The workflow of the image variation latent diffusion models is shown in Figure 1. The latent diffusion models include three main stages: encoding, denoising, and decoding. The encoding process has two key components: the Encoder and Quantization layers. Similarly, the decoding process contains the Decoder and Post-Quantization layers. The denoising process is in a Unet structure containing Resnet, Self-Attention, Cross-Attention, and Feed-Forward modules. In this paper, we

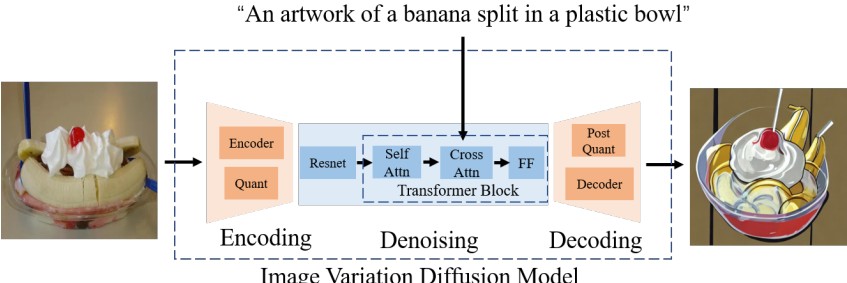

"An artwork of a banana split in a plastic bowl"

Figure 1: The workflow of the image variation latent diffusion model for image editing.

aim to analyze the adversarial robustness of these modules. The workflow of image inpainting models is shown in Figure 5 in the Appendix.

## 2.2 ADVERSARIAL ATTACKS

Given the target image $x$ and a DNN model $f(x)$, adversarial attacks aim to find an adversarial sample $x^{adv}$, which can mislead the DNN model, while it is human-imperceptible, i.e., satisfying the constraint $\left\| x - x^{adv} \right\|_p < \epsilon$. $\left\| \cdot \right\|_p$ represents the $L_p$ norm, and we focus on the $L_\infty$ norm in this paper to align with previous adversarial attack research Dong et al. (2018); Lin et al. (2019).

Prevailing adversarial attacks like FGSM Goodfellow et al. (2014) usually craft adversarial samples by solving the following constrained maximization problem for attacking an image classifier with the cross entropy loss $J(x, y)$, where $y$ is the ground truth label for the image $x$:

$$\max_{x^{adv}} \quad J(x^{adv}, y) \quad s.t. \left\| x - x^{adv} \right\|_\infty < \epsilon.$$

However, in reality, the model structure and weights are hidden from the users. Therefore, researchers focus on the adversarial attack under the black-box setting. Among different black-box attack methodologies, transfer-based attacks stand out due to their severe security threat to deep learning-based applications in practice. Therefore, we also examine the transfer-based attacks against diffusion models in this paper. Specifically, transfer-based attacks Zhang et al. (2023) usually craft adversarial samples with a local source model using white-box attack methods and directly employ the generated adversarial samples to mislead the target model. There are also some works Xue et al. (2023); Chen et al. (2023) generate adversarial examples with the help of diffusion models.

## 2.3 ADVERSARIAL ATTACKS ON DIFFUSION MODELS

Current adversarial attacks on diffusion models mainly focus on the white-box scenario. Zhuang et al. (2023) attacks diffusion models by adding extra meaningless characters to the input prompt to mislead the text encoder. Salman et al. (2023); Liang & Wu (2023) tries to attack the image encoder and distort the output image to disable the normal functionality of latent diffusion models. Furthermore, some works Liang et al. (2023); Shan et al. (2023) also utilize adversarial examples to prevent privacy issues. However, previous works do not explore the core of the diffusion models (i.e., the denoising process). Besides, current research only considers the white-box situation and neglects the black-box setting, which is closer to the real-world setting. In this paper, we more thoroughly analyze the vulnerability of the modules inside diffusion models under the white-box scenario. We also examine the attacking performance under the transfer-based black-box setting, which covers both the prompt-transfer and model-transfer situations.

## 3 METHODOLOGY

In this section, We evaluate the robustness of latent diffusion models from the perspective of adversarial attacks. We first state the problem of adversarial attacks on latent diffusion models. Then, we explain our attacking objectives.

## 3.1 PROBLEM STATEMENT

Given the input image $x$ and a prompt $p$, the latent diffusion model $f(\cdot)$ generates the edited image $f(x, p)$ with the guidance of $p$. We aim to find the adversarial image $x^{adv}$ of the latent diffusion model, such that the adversarial image cannot be edited following the guidance of $p$. Additionally, $x^{adv}$ satisfies the constraint $\|x - x^{adv}\|_\infty < \epsilon$ by following previous works Dong et al. (2018); Lin et al. (2019).

Unlike the previous adversarial attacks on image classifiers, it is hard to quantify the effectiveness of adversarial attacks on diffusion models with an explicit objective function. The generative model's output is usually evaluated by the human judgment $H(\cdot)$, which determines the similarity between the edited image and the prompt. Therefore, attacking the latent diffusion model is to minimize $H(f(x, p), p)$, which can be formalized as follows:

$$\min_{x^{adv}} \; H(f(x^{adv}, p), p) \quad s.t. \left\|x - x^{adv}\right\|_\infty < \epsilon. \tag{1}$$

## 3.2 ATTACKING METHOD

However, the human judgment $H(\cdot)$ is a qualitative measurement, and we cannot directly minimize $H(\cdot)$. Therefore, we instead perturb the intermediate features inside diffusion models. If the features inside the diffusion models are largely perturbed, the original functionality of the diffusion models is also expected to be largely destroyed, based on the assumptions from feature-level adversarial attacks Zhang et al. (2022). Attacking the hidden representations also sheds light on analyzing the vulnerability of each component inside diffusion models.

As a result, we maximize the $l_2$ distance between the intermediate representations of the adversarial example and the original one inside the latent diffusion model. As we discussed in the related work, we propose to separately attack the three stages of latent diffusion models and their components. The attacking objective is formulated as:

$$\max_{x^{adv}} \; \left\|f^m(x, p) - f^m(x^{adv}, p)\right\|_2 \quad s.t. \left\|x - x^{adv}\right\|_\infty < \epsilon, \tag{2}$$

where $f^m(\cdot)$ represents the output of the module $m$ inside the model $f$ given the input image $x$ and the prompt $p$.

## 3.3 ATTACKING MODULES

We propose to attack all important modules inside the model individually to comprehensively analyze the robustness of latent diffusion models. The image editing latent diffusion models contain three crucial processes: encoding, denoising, and decoding, as shown in Figure 1. The encoding process first utilizes a variational autoencoder to encode the image and a vector quantization layer to regularize the latent space. We select the image Encoder and the Quantization layer as the attacking modules for the encoding part. Then, the Unet denoises the latent space and combines the information from the prompt step by step. In each step, the Unet consists of two basic components: the Resnet block and the transformer block, which compute the latent features and combine the information from the prompt, respectively. Especially, the transformer block includes Self-Attention, Cross-Attention, and Feed-Forward layers. Therefore, we select Resnet, Self-Attention, Cross-Attention, and Feed-Forward layers (FF) for adversarial attacks. Finally, the latent features are decoded to the output image through a Post-Quantization layer and the Decoder part of the variational autoencoder, so we choose to attack the Post-Quantization layer and the Decoder.

## 3.4 DATASET CONSTRUCTION

In the literature, there is no publicly available dataset for adversarial attacks on diffusion models. Therefore, in this section, we present our automatic dataset construction pipeline for building the dataset. We first state the data source for dataset construction and illustrate the detailed steps to build the dataset. The whole process is automatic, as shown in Figure 2 and Figure 6, and human is only required to further guarantee the quality of the dataset at the last step. Please note that our dataset is not only available for evaluating image editing diffusion models. The generated prompts can also be utilized to evaluate the robustness of text2image diffusion models. More details are shown in the Appendix.

### 3.4.1 DATA SOURCE

To align with the experimental setting of the diffusion model evaluation Rombach et al. (2022), we select the validation set of the coco dataset Lin et al. (2014) as the data source for constructing the dataset. The coco validation dataset contains 5000 images sampled from the natural world with 91 classes, which is complex and diverse enough to test the robustness of diffusion models. Furthermore, coco is the most fundamental dataset for a variety of computer vision tasks, including classification, segmentation, and caption, providing abundant information to fulfill the dataset requirement, e.g., modifying the caption to the image editing prompt or extracting the mask for an object.

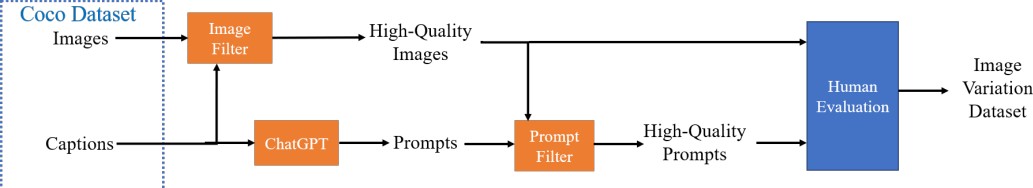

Figure 2: The automatic dataset construction pipeline for image variation diffusion model.

### 3.4.2 CONSTRUCTION PIPELINE

We propose two different pipelines for the adversarial attack on the image editing functionality of diffusion models based on the model category. One pipeline generates input pairs (image, prompt) for the image variation diffusion model in Figure 2, and another pipeline produces input triplets (image, prompt, mask) for the image inpainting diffusion models in Figure 6 in the Appendix.

**Pipeline Image variation**

The pipeline generates pairs of images and text prompts. Since the image caption task of the coco dataset consists of 5 human descriptions per image, we can modify each description a little bit to a text prompt without changing the main entity. Therefore, we can directly utilize the images from the coco dataset and change the corresponding captions to form the image variation dataset.

**Step 1: Data Preprocessing.** The images from the coco dataset are sampled from the real world, but the contents in the image are complex, and sometimes the main entity is not clear. With the aim of selecting the images with clear main entities, we first rank the images based on the average CLIP score Radford et al. (2021) between the image and each of the image captions. 10% of the images are selected to form the dataset. Afterward, we select the top 3 captions for each image based on the CLIP score to guarantee the high-quality of image captions. As a result, we obtain 500 high-quality images and three high-quality captions per image.

**Step 2: Prompt Generation.** In order to automatically modify the image caption to the text prompt, we deploy the strong power of the large language model Ouyang et al. (2022); Brown et al. (2020) for modifying the captions. We query the ChatGPT (GPT 3.5 version) by the following question "Please modify the following sentence ¡xxx¿ to generate 5 similar scenes without changing the entities." to generate 5 text prompts per image caption. For the purpose of successful image editing, we rank and select the top-5 text prompts by the CLIP score between the generated prompt and the output image of Stable Diffusion V1-5 Rombach et al. (2022). Finally, we obtain five high-quality text prompts per image.

**Extra Step: Human Evaluation.** A human volunteer is asked to rank the data pair based on the visual performance of the generated images and the coherence between the generated image and the prompt. Finally, the top-100 images are selected, and there are 100*5=500 data pairs for the dataset.

## 4 EXPERIMENT

In this section, we conduct extensive experiments to analyze the robustness of latent diffusion models under different experiment settings on the constructed dataset. We first specify the setup of the experiments. Then, we present the white-box attacking results under different experiment settings.

Table 1: The attacking performance against different modules inside SD-v1-5 image variation model by various measurement. The best results are marked in bold.

| Module | CLIP | PSNR | SSIM | MSSSIM | FID | IS |
|---|---|---|---|---|---|---|
| Encoder | 33.82 | 15.58 | 0.226 | 0.485 | 172.5 | 15.05 |
| Quant | 34.54 | 16.19 | 0.250 | 0.533 | 168.2 | 15.77 |
| Resnet | **29.89** | **11.82** | **0.076** | **0.270** | **206.3** | 16.93 |
| Self Attn | 32.37 | 14.91 | 0.108 | 0.305 | 206.2 | **12.19** |
| Cross Attn | 34.48 | 16.22 | 0.250 | 0.557 | 171.3 | 15.24 |
| FF | 31.72 | 13.49 | 0.096 | 0.290 | 190.1 | 17.33 |
| Post Quant | 33.17 | 13.39 | 0.169 | 0.402 | 202.8 | 17.0 |
| Decoder | 34.14 | 15.05 | 0.223 | 0.509 | 184.0 | 20.59 |
| Gaussian | 34.18 | 15.49 | 0.198 | 0.510 | 179.0 | 15.41 |
| Benign | 34.74 | $\infty$ | 1.000 | 1.000 | 167.9 | 19.86 |

Table 2: The white-box attacking performance against different image variation diffusion models by various measurement. The best results are marked in bold.

| Model | Module | CLIP | PSNR | SSIM | MSSSIM | FID | IS |
|---|---|---|---|---|---|---|---|
| SD-v1-4 | Encoding | 34.09 | 15.47 | 0.225 | 0.485 | 172.7 | 15.8 |
| | Denoising | **30.93** | **12.23** | **0.080** | **0.288** | **193.4** | 19.8 |
| | Decoding | 33.00 | 13.30 | 0.168 | 0.405 | 190.1 | **14.5** |
| | Gaussian | 34.05 | 15.42 | 0.199 | 0.509 | 174.4 | 19.8 |
| | Benign | 34.54 | $\infty$ | 1.000 | 1.000 | 174.5 | 19.8 |
| SD-v1-5 | Encoding | 33.82 | 15.58 | 0.226 | 0.485 | 172.5 | **15.05** |
| | Denoising | **29.89** | **11.82** | **0.076** | **0.270** | **206.3** | 16.93 |
| | Decoding | 33.17 | 13.39 | 0.169 | 0.402 | 202.8 | 17.00 |
| | Gaussian | 34.18 | 15.49 | 0.198 | 0.510 | 179.0 | 15.41 |
| | Benign | 34.74 | $\infty$ | 1.000 | 1.000 | 167.9 | 19.86 |
| SD-v2-1 | Encoding | 30.62 | 14.13 | 0.156 | 0.392 | 217.8 | **10.39** |
| | Denoising | 27.98 | **12.25** | **0.111** | **0.314** | **217.4** | 13.57 |
| | Decoding | **27.61** | 12.30 | **0.111** | 0.317 | 212.2 | 13.93 |
| | Gaussian | 31.77 | 13.70 | 0.156 | 0.425 | 202.6 | 14.68 |
| | Benign | 32.03 | $\infty$ | 1.000 | 1.000 | 201.8 | 13.74 |
| Instruct | Encoding | 33.94 | 9.89 | **0.125** | **0.338** | **180.9** | **16.06** |
| | Denoising | **31.65** | 11.87 | 0.153 | 0.399 | 175.8 | 17.46 |
| | Decoding | 33.75 | **9.81** | 0.146 | 0.364 | 167.9 | 20.86 |
| | Gaussian | 33.98 | 12.13 | 0.197 | 0.457 | 167.9 | 20.86 |
| | Benign | 34.45 | $\infty$ | 1.000 | 1.000 | 153.75 | 22.89 |

Besides, we demonstrate the black-box attacking performance under two transfer settings. Finally, we illustrate the robustness of the adversarial examples.

## 4.1 EXPERIMENTAL SETUP

**Target Model**. We consider both image variation and image inpainting models as the target models. For image variation models, we choose four widely used models, containing Stable Diffusion V1-4 (SD-v1-4) Rombach et al. (2022), Stable Diffusion V1-5 (SD-v1-5), Stable Diffusion V2-1 (SD-v2-1), and Instruct-pix2pix (Instruct) Brooks et al. (2022). Different versions of diffusion models have the same structure. The higher version is further trained based on the previous version. While, Instruct-pix2pix has a different structure with Stable Diffusion Models. For image inpainting models, we consider two models: Stable Diffusion v1-5 and Stable Diffusion v2-1. Besides, we include three input-level defense methods that are robust against adversarial attacks. These defenses cover random resizing and padding (R&P) Xie et al. (2017), JPEG compression Liu et al. (2019), and Gaussian noise (Gaussian) Li et al. (2019).

**Metric**. We evaluate the performance of adversarial attacks on diffusion models from two perspectives: the quality of the generated image and the functionality of image editing. The quality of generated image are measured via Fréchet Inception Distance (FID) Heusel et al. (2017) and Inception Score (IS) Salimans et al. (2016). The evaluations of image editing functionality are categorized into two levels: image and text. The image level measures the similarity of generated images between

Table 3: The prompt-transfer attacking performance on SD-v1-5 image variation model by various measurement. The best results are marked in bold.

| Module | CLIP | PSNR | SSIM | MSSSIM | FID | IS |
|---|---|---|---|---|---|---|
| Encoding | 33.82 | 15.58 | 0.226 | 0.485 | 172.5 | **15.05** |
| Denoising | **28.27** | **11.92** | **0.074** | **0.271** | 183.33 | 24.00 |
| Decoding | 30.18 | 13.55 | 0.166 | 0.400 | **187.85** | 20.56 |
| Gaussian | 34.18 | 15.49 | 0.198 | 0.510 | 179.0 | 15.41 |

the adversarial examples and benign ones. We select Peak-Signal-to-Noise Ratio (PSNR), Structural Similarity Index Measure (SSIM) Wang et al. (2004), and Multi-Scale Structural Similarity Index Measure (MSSSIM) Wang et al. (2003) as the similarity metrics. The text level functionality measures the similarity between the generated image and text prompt by the CLIP score Radford et al. (2021). In general, image level similarity measures the normal function, reflecting the normal execution given the input image and edited prompt. Similarly, text-level similarity represents the expected function, representing the expected execution of the LDM given the edited prompt.

**Parameter**. All the experiments are conducted on an A100 GPU server. Following Kurakin et al. (2016), we set the maximum perturbation budget $\epsilon = 0.1$, the number of attack iterations $T = 15$, and the step length $\epsilon' = 0.01$. The number of the diffusion step is set to be 15 for attack and 100 for inference. We take strength and guidance of the diffusion models to be 0.7 and 7.5 by default.

## 4.2 WHITE-BOX PERFORMANCE

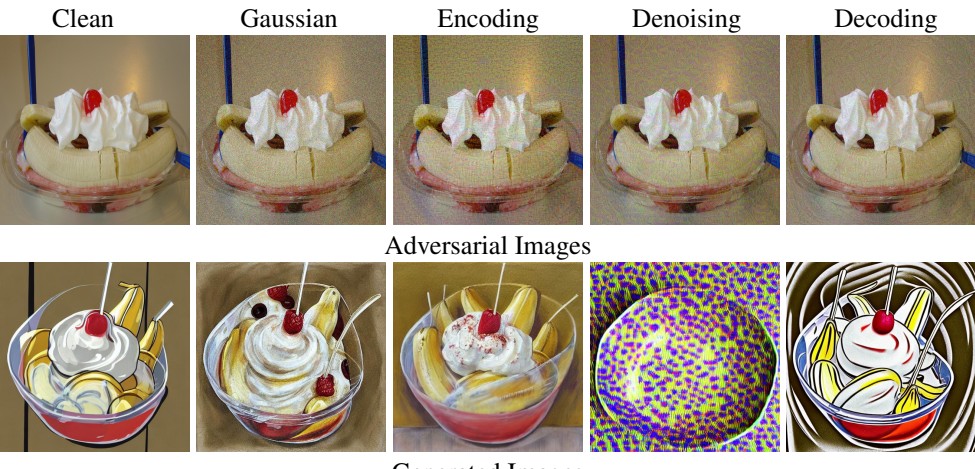

Figure 3: Visualization of adversarial images and their output of the SD-v1-5 image variation model with the prompt "An artwork of a banana split in a plastic bowl".

We first illustrate the white-box attacking performance on different modules inside the Stable Diffusion V1-5 image variation and inpainting models, respectively. Then we select the most vulnerable module for each process and evaluate the white-box performance on other diffusion models.

As shown in Table 1, we analyze the attacking performance on different modules of the SD-v1-5 image variation model. We first analyze the expected function that the CLIP scores of adversarial attacks are all lower than the Gaussian noise, validating that the diffusion models are susceptible to adversarial attacks. In addition, attacking the denoising process achieves the lowest CLIP score compared with attacking the encoding or decoding process. Astonishingly, the CLIP score is reduced to 29.89 with a marginal drop of 4.85, destroying the expected function severely. From the point of view of both normal function and image quality, attacking the denoising process and especially the Resnet module outperforms attacking other processes and modules. We also observe that attacking the cross attention performs badly. We think the reason is the prompt information dominates the cross attention module, so attacking the image is not effective, which requires textual adversarial attacks to corrupt this module. The qualitative results are shown in Figure 3. In general,

Table 4: The model-transfer attacking performance against different image variation diffusion models. The best results are marked in bold.

| Model | Module | SD-v1-4 | SD-v1-5 | SD-v2-1 | Instruct |
|-------|--------|---------|---------|---------|----------|
| SD-v1-4 | Encoding | 34.09 | 33.82 | 30.60 | 33.94 |
| | Denoising | **30.93** | **31.40** | **29.13** | **33.70** |
| | Decoding | 33.00 | 33.07 | 29.71 | 33.86 |
| SD-v1-5 | Encoding | 34.11 | 33.82 | 30.65 | 33.94 |
| | Denoising | **29.73** | **29.89** | **28.48** | **33.30** |
| | Decoding | 33.15 | 33.17 | 29.47 | 33.93 |
| SD-v2-1 | Encoding | 34.09 | **33.84** | 30.62 | 33.93 |
| | Denoising | **33.85** | 33.94 | 27.98 | **33.90** |
| | Decoding | 33.88 | 34.07 | **27.61** | 34.03 |
| Instruct | Encoding | **33.96** | 34.07 | 31.24 | 33.94 |
| | Denoising | 34.08 | **34.02** | **30.52** | **31.65** |
| | Decoding | 34.23 | 34.22 | 31.74 | 33.75 |

the generated images by the adversarial examples are of low visual quality, and the editing is meaningless. Furthermore, the qualitative results also validate our conclusion that attacking the denoising process is effective. We can achieve a similar conclusion on attacking image inpainting diffusion models in the Appendix.

In addition to the white-box analysis on Stable Diffusion v1-5, we illustrate the attacking performance on other diffusion models as shown in Table 2. We select to attack the most vulnerable modules of each process and measure the attacking performance. We can see all the diffusion models are susceptible to adversarial attacks, which can largely influence the normal function and image quality. The Instruct-pix2pix is more robust compared with standard stable diffusion models with higher CLIP score and similarity with the benign output. Attacking the Denoising is consistently effective compared with attacking other processes of the image variation diffusion models. More white-box performance of the image inpainting model is shown in the Appendix.

| SD-v1-4 | SD-v2-1 | R & P | JPEG |
|---------|---------|-------|------|

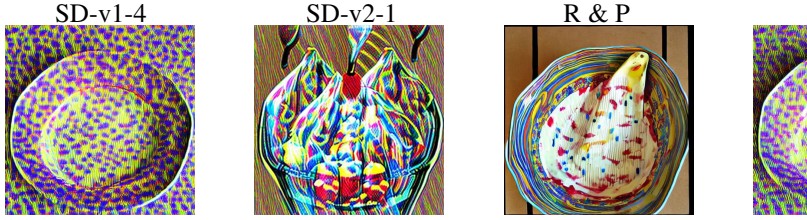

Figure 4: Visualization of generated images by model-transfer attacks on SD-v1-4 and SD-v2-1, and generated images of the SD-v1-5 against defense mechanisms with the prompt "An artwork of a banana split in a plastic bowl".

### 4.3 BLACK-BOX PERFORMANCE

In this section, we demonstrate the black-box performance, especially, the transfer-based attacks. Unlike the transfer-based attacks for image classifiers, we categorize the transfer-based attacks for diffusion models into two classes: model-transfer and prompt-transfer. Model-transfer is similar to transfer-based attacks on image classifiers in that the generated adversarial examples by one diffusion model can mislead the other diffusion models. Prompt-transfer is from the perspective of data that the generated adversarial examples can also influence the diffusion model with other similar prompts.

We first illustrate the prompt-transfer performance on the Stable Diffusion V1-5 image variation model. We craft the adversarial image under one prompt and transfer the crafted image combined with other similar prompts to evaluate the generative results. As shown in Table 3, the prompt-transfer achieves similar results on CLIP score and normal function with the original prompt. The adversarial images crafted by one prompt can also mislead the guidance of other similar prompts. Therefore, we can successfully mislead the diffusion models by adversarial samples without knowing the exact text prompts.

Table 5: The attacking performance against SD-v1-5 image variation model by various defense mechanisms. The best results are marked in bold.

| Methods | Module | CLIP | PSNR | SSIM | MSSSIM | FID | IS |
|---|---|---|---|---|---|---|---|
| R & P | Encoding | 33.94 | 12.42 | 0.187 | 0.413 | 179.0 | 24.36 |
| | Denoising | **33.84** | **12.20** | **0.173** | **0.397** | **181.0** | **23.64** |
| | Decoding | 33.92 | 12.30 | 0.190 | 0.418 | 177.8 | 24.90 |
| JPEG | Encoding | 34.11 | 15.94 | 0.252 | 0.513 | 196.0 | 16.02 |
| | Denoising | **30.75** | **12.23** | **0.087** | **0.299** | **234.8** | **9.74** |
| | Decoding | 33.50 | 14.00 | 0.195 | 0.437 | 189.6 | 19.02 |
| Gaussian | Encoding | 33.84 | 14.58 | 0.140 | 0.441 | 170.3 | **22.04** |
| | Denoising | **30.75** | **12.06** | **0.072** | **0.289** | **183.1** | 22.69 |
| | Decoding | 33.39 | 13.50 | 0.127 | 0.401 | 173.4 | 22.94 |

Then, we consider the model-transfer performance that we craft adversarial examples on the victim model and directly transfer the adversarial samples to other diffusion models. As shown in Table 4, the transfer attacks can mislead the function because the CLIP scores are all reduced compared with the benign results in Table 2. Instruct-pix2pix is hard to transfer to other standard diffusion models, and other diffusion models are also hard to transfer to Instruct-pix2pix because of the different model architectures. Considering the transfer attacks between different versions of standard stable diffusion models, adversarial examples crafted by SD-v1-4 and SD-v1-5 are easy to mislead SD-v2-1. The experiment result means the SD-v2-1 is more vulnerable, and the defects inside the previous versions of stable diffusion models are inherited by SD-v2-1. Furthermore, adversarial samples from SD-v2-1 cannot largely mislead SD-v1-4 or SD-v1-5, representing the deficiencies of SD-v2-1 are not the common problems for SD-v1-4 or SD-v1-5. Combining the two observations, we conclude the problems of SD-v1 are inherited by SD-v2, and SD-v2 has more defects compared with SD-v1. With the updating of the stable diffusion models, models become more vulnerable and have more defects, which raises the alarm about the robustness of diffusion models. The researchers should also consider the robustness issue when they update the version of diffusion models. The qualitative results are shown in Figure 4. We can see that transfer-based attacks can mislead the functionality of other diffusion models, raising concerns about the robustness of diffusion models. More model-transfer results on image inpainting diffusion models are shown in the Appendix.

## 4.4 Robustness of Adversarial Examples

We further analyze the robustness of adversarial examples because the robustness of the adversarial examples has practical issues. If the adversarial images are robust, we can deploy them to avoid the image editing model's misuse in real life. We consider three kinds of defense mechanisms on the image to mitigate the adversarial influence, including geometry transformation, compression, and noise. Especially, We select Random Resizing and Padding (R & P), JPEG compression (JPEG), and Gaussian noise (Gaussian) to evaluate the robustness of adversarial examples. All the defense mechanisms can mitigate the adversarial influence on the normal function and expected function as shown in Table 5. Significantly, the geometry transformation (R & P) can largely alleviate the adversarial problems with a 3.95 increment of the CLIP score on attacking the Denoising. The qualitative results are also shown in Figure 4. Although the defense mechanisms can mitigate the adversarial issues, the expected function is still far from satisfaction in the qualitative examples.

## 5 Conclusion

In this paper, we explore the robustness of the diffusion models from the perspective of adversarial attacks. We first demonstrate that the denoising process, especially the Resnet module, is the most vulnerable component in the diffusion models. We further consider two transfer-based black-box scenarios: prompt-transfer and model-transfer. The adversarial images can transfer well between prompts and models. Significantly, we figure out that SD-v2 is more vulnerable than SD-v1, and defects inside SD-v1 are inherited by SD-v2. This observation raises concerns about the robustness of the diffusion model development. Additionally, we propose automatic dataset construction pipelines for building a high-quality publicly available dataset to complement the literature.

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

APPENDIX

In this appendix, Section A illustrates the workflow of image inpainting models. Section B demonstrates the dataset construction pipeline for image inpainting models. Section C shows the detailed experimental setting for each experiment in the paper. Section D illustrates the constructed dataset. Section E reports the attacking performance against latent diffusion image inpainting models under the white-box and black-box scenarios. Section F demonstrates more qualitative results on different attacking scenarios. Finally, we discuss several aspects of the adversarial attacks on latent diffusion models, including the usage, efficiency issues, and limitations in Section G.

## A  WORKFLOW OF IMAGE INPAINTING MODELS

The workflow of image inpainting model is in Figure5, which is similar to image variation model.

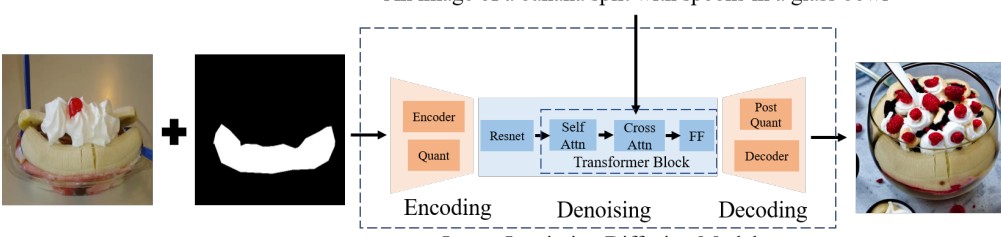

Figure 5: The workflow of image inpainting latent diffusion model for image editing.

## B  DATASET CONSTRUCTION PIPELINE

**Pipeline Image Inpainting**

Image inpainting pipeline generates triplets of image, prompt and mask. The mask covers the region of the main entity, and the diffusion models change the remaining part of the image with the guidance of the prompt. We utilize the bounding box in the detection task to discover the main entity of the image and employ the segmentation task to directly achieve the mask of the main entity. The detailed steps for the dataset construction are followed.

**Step 1: Data Preprocessing.** The same to pipeline image variation.

**Step 2: Prompt Generation.** The same to pipeline image variation.

**Step 3: Main Entity Finder.** We assume the size of the main entity as well as the similarity between the main entity and the image should be large. Based on the first assumption, the top-5 large objects in the image have the potential to be the main entity. Then, We select the object with the highest CLIP score between the image and each category of the potential object to be the main entity. Besides, if other top-5 objects have the same category as the main entity, we consider there are multiple main entities inside the image, and we take the union of their masks.

**Step 4: Image Cropper.** To avoid the diffusion models being unaware of the main entity, we should guarantee the main entity object takes a large region in the image. Therefore, we adaptively center-crop the image based on the size of the main entity.

**Extra Step: Human Evaluation.** The same to pipeline image variation.

## C  EXPERIMENTAL SETTING

In this section, we further clarify the experimental settings for each experiment in the paper.

There are, in total, six metrics to measure the functionality of the latent diffusion models from three aspects. CLIP score measures the similarity between the generated image and the edited prompt

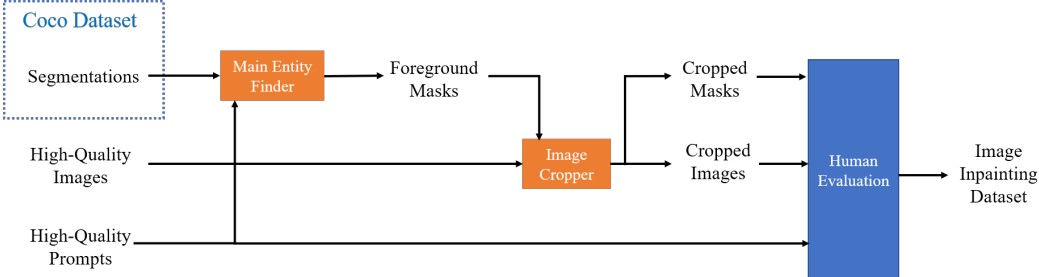

Figure 6: The automatic dataset construction pipeline for image inpainting diffusion model.

representing the expected function. A low CLIP score means the generated image is far from the expected editing of the prompt. The normal function disruption is measured by three metrics: PSNR, SSIM, and MSSSIM. They compute the similarity between the benign input and the corresponding adversarial input, and a low score reflects a large normal function disruption. The quality of the generated image is measured by FID and IS. A lower FID and a higher IS indicate that the generated images are of higher quality.

We select the most well-known latent diffusion models: Stable Diffusion v1-5 for the white-box component experiments. We consider eight different modules inside the SD-v1-5, as explained in Section 3.3 of the paper. We deploy the adversarial attacks described in Equation (2) on the target model's output to destroy the intermediate features of the target module. Since there are multiple denoising steps, we consider to destroy the target module inside the denoising process on all the steps. We also select Gaussian noises as the standard baseline. The experimental results show that the adversarial noises are better than the Gaussian noise.

For the white-box experiment on other latent diffusion models, We select to attack the most vulnerable modules of each process. Specifically, We choose the Encoder module for the Encoding process, Resnet for denoising process, and Post Quantization layers for the Decoding process.

We consider two kinds of transfer-based attack scenarios in diffusion attacks. The prompt-transfer means that the adversarial image crafted by one prompt can also mislead the functionality of the diffusion model with a similarly edited prompt input. We test the prompt-transfer in a circulation manner in the experiment. Specifically, we craft one adversarial image by the current prompt and test the prompt-transfer by the next prompt of the image when we fix the image and model. We also evaluate the model-transfer and directly transfer the adversarial image by fixing the image and prompt.

## D   DATASET ILLUSTRATIONS

The dataset contains two subsets for different categories of image editing diffusion models. Each subset includes 100 images and five edited prompts per image. The subset for image inpainting diffusion models also has additional 100 image masks for the images. The qualitative results of the dataset are shown in Figure 7 and Figure 8. We will release the whole dataset and the code for constructing the dataset to boost the research on the robustness of diffusion models.

## E   ATTACKING PERFORMANCE

We present more results on attacking latent diffusion image inpainting models in this section. We analyze the attacking performance on different modules of the SD-v1-5 image inpainting model and the attacking performance on the selected image inpainting latent diffusion models under the white-box scenario. Besides, we evaluate the model-transfer performance across different versions of the Stable Diffusion models.

We show the attacking performance on different modules of the SD-v1-5 inpainting model in Table 6. Compared with attacking image variation models, it is challenging to mislead its functionality, since the output image keeps the main entity, and the editing region is limited. However, adversarial

Images                                        Prompts

"A depiction of spoons and whipped cream on a classic banana split served in a traditional way."
"An artwork of a banana split in a plastic bowl."
"A depiction of a banana split in a plastic bowl on a canvas."
"An image of a banana split with spoons in a glass bowl."
"An image of a traditional banana split with whipped cream and spoons."

"Sound asleep, the cat is curled up in a circular tin."
"Curled up and dozing off, the cat has made a round tin its new bed."
"The sight of a cat sleeping soundly in a bowl on the ground is enough to make anyone feel cozy."
"A bowl on the ground is transformed into a cozy bed for a cat catching some z's."
"A bowl on the ground serves as the perfect bed for a sleepy cat taking a nap."

Figure 7: The data pair examples of the constructed dataset for the image variation diffusion models.

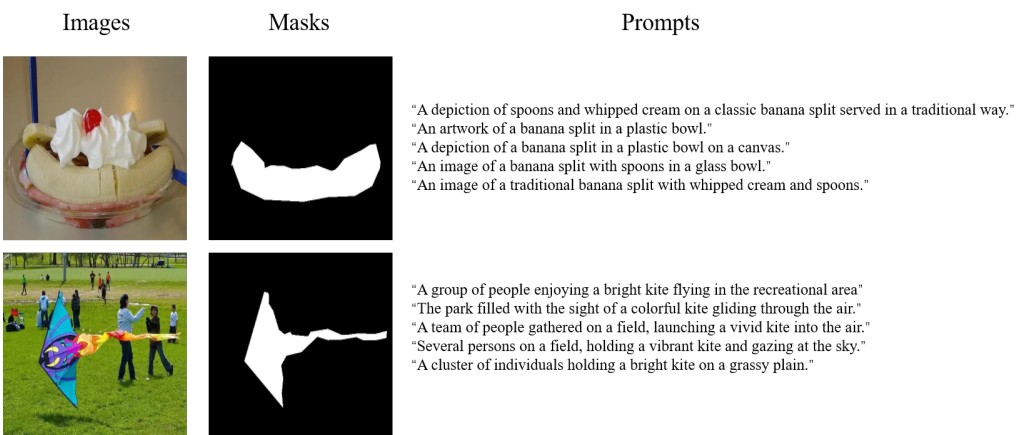

Images                  Masks                          Prompts

"A depiction of spoons and whipped cream on a classic banana split served in a traditional way."
"An artwork of a banana split in a plastic bowl."
"A depiction of a banana split in a plastic bowl on a canvas."
"An image of a banana split with spoons in a glass bowl."
"An image of a traditional banana split with whipped cream and spoons."

"A group of people enjoying a bright kite flying in the recreational area"
"The park filled with the sight of a colorful kite gliding through the air."
"A team of people gathered on a field, launching a vivid kite into the air."
"Several persons on a field, holding a vibrant kite and gazing at the sky."
"A cluster of individuals holding a bright kite on a grassy plain."

Figure 8: The data triplet examples of the constructed dataset for the image inpainting diffusion models.

attacks can still reduce the CLIP score by more than 1.3, and the image quality as well as the normal function are influenced. The qualitative results are shown in Figure 9. Among the modules, attacking the Resnet achieves the best performance on destroying the normal function and the image quality, but it is a little bit inferior than attacking the Encoder. Since an image inpainting model has the mask input, and the edited region is limited, attacking the Denoising may overfit the model, leading to a slightly lower expected function disruption and a higher normal function disruption compared with attacking the Encoder. Therefore, Resnet and Encoder are more vulnerable than other modules for image inpainting diffusion models.

Additionally, as shown in Table 7, we can draw the same conclusion as the other image inpainting latent diffusion models that the encoding and denoising are vulnerable. Attacking the encoding process can disrupt the expected function, while destroying the denoising is able to mislead the normal function and reduce the image quality, which can be validated from the qualitative results.

Besides, we illustrate the model-transfer performance in Table 8. We can see that the transfer attack can achieve better performance than the Gaussian noise in Table 2. Therefore, the adversarial examples of image inpainting models still transfer across different models. Furthermore, the adversarial images crafted from SD-v1-5 and SD-v2-1 can achieve similar performance on attacking SD-v2-1, which means that the adversarial examples from SD-v1-5 are as effective as the ones from SD-v2-1 on misleading SD-v2-1. However, the adversarial examples crafted from SD-v2-1 cannot mislead SD-v1-5 well. Thus, the image inpainting latent diffusion models suffer from the same robustness problem as image variation models. The defects from SD-v1 are inherited by SD-v2, which also includes new defects.

Table 6: The attacking performance against different modules inside the SD-v1-5 image inpainting model. The best results are marked in bold.

| Module | CLIP | PSNR | SSIM | MSSSIM | FID | IS |
|--------|------|------|------|--------|-----|-----|
| Encoder | **33.04** | 14.33 | 0.255 | 0.538 | **174.8** | 17.69 |
| Quant | 34.05 | 14.83 | 0.271 | 0.572 | 155.6 | 24.25 |
| Resnet | 33.44 | 13.53 | **0.219** | **0.495** | 172.3 | **14.82** |
| Self Attn | 33.61 | 13.89 | 0.220 | 0.500 | 162.2 | 18.99 |
| Cross Attn | 33.50 | 14.82 | 0.275 | 0.572 | 154.6 | 23.93 |
| FF | 33.57 | 14.27 | 0.221 | 0.500 | 162.4 | 20.97 |
| Post Quant | 33.69 | **13.29** | 0.248 | 0.520 | 166.3 | 21.65 |
| Decoder | 34.17 | 15.07 | 0.282 | 0.594 | 155.7 | 24.73 |
| Gaussian | 34.31 | 15.95 | 0.316 | 0.628 | 161.1 | 19.78 |
| Benign | 34.4 | $\infty$ | 1.000 | 1.000 | 156.4 | 22.45 |

Table 7: The white-box attacking performance against different image inpainting diffusion models. The best results are marked in bold.

| Model | Module | CLIP | PSNR | SSIM | MSSSIM | FID | IS |
|-------|--------|------|------|------|--------|-----|-----|
| | Encoding | **33.04** | 14.33 | 0.255 | 0.538 | **174.8** | 17.69 |
| | Denoising | 33.44 | 13.53 | **0.219** | **0.495** | 172.3 | **14.82** |
| SD-v1-5 | Decoding | 33.69 | **13.29** | 0.248 | 0.520 | 166.3 | 21.65 |
| | Gaussian | 34.31 | 15.95 | 0.316 | 0.628 | 161.1 | 15.78 |
| | Benign | 34.40 | $\infty$ | 1.000 | 1.000 | 156.4 | 22.45 |
| | Encoding | **33.62** | 14.24 | 0.268 | 0.542 | **184.3** | 13.74 |
| | Denoising | 33.98 | **13.75** | **0.252** | **0.524** | 174.1 | 17.83 |
| SD-v2-1 | Decoding | 34.20 | 13.86 | 0.270 | 0.542 | 168.7 | 19.00 |
| | Gaussian | 34.66 | 15.97 | 0.334 | 0.635 | 155.7 | 22.88 |
| | Benign | 34.88 | $\infty$ | 1.000 | 1.000 | 153.0 | 23.56 |

Table 8: The model-transfer attacking performance against other image inpainting diffusion models. The best results are marked in bold.

| Model | Module | SD-v1-5 | SD-v2-1 |
|-------|--------|---------|---------|
| | Encoding | **33.04** | **33.61** |
| SD-v1-5 | Denoising | 33.44 | 33.87 |
| | Decoding | 33.69 | 34.31 |
| | Encoding | **33.61** | **33.62** |
| SD-v2-1 | Denoising | 33.72 | 33.98 |
| | Decoding | 33.87 | 34.20 |

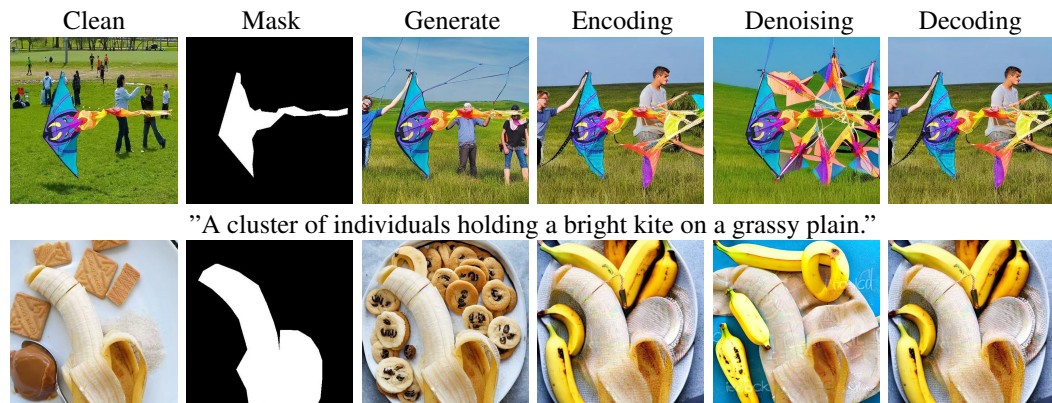

| Clean | Mask | Generate | Encoding | Denoising | Decoding |

"A cluster of individuals holding a bright kite on a grassy plain."

"A platter with a banana that is sliced, cookies, and a spoon resting on it."

Figure 9: Visualization of adversarial images and their output of the SD-v1-5 image inpainting model.

# F QUALITATIVE RESULTS

We add more white-box qualitative results against the SD-v1-5 image variation model in Figure 10. We also include the corresponding black-box qualitative results and the results against defense mechanisms in Figure 11.

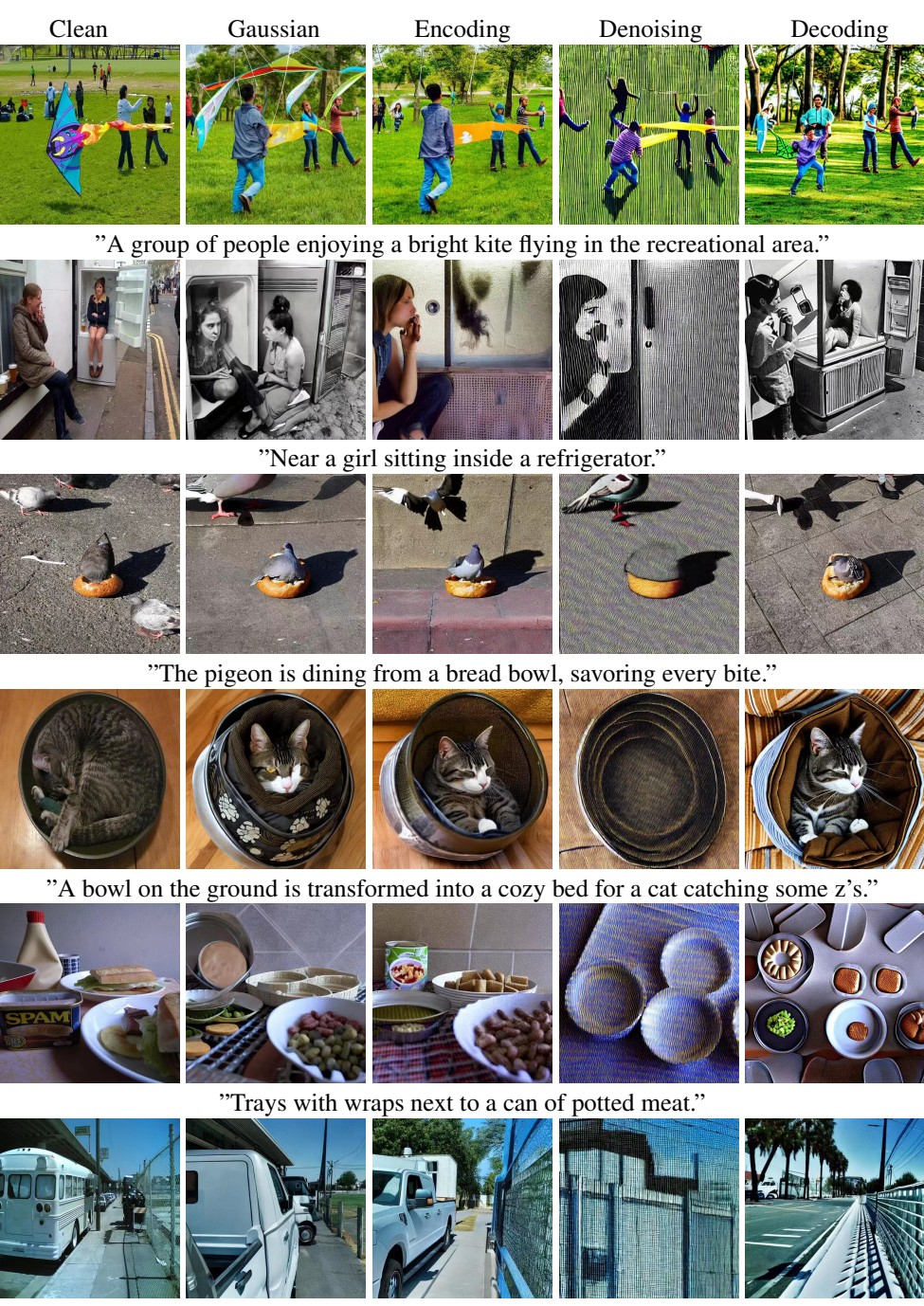

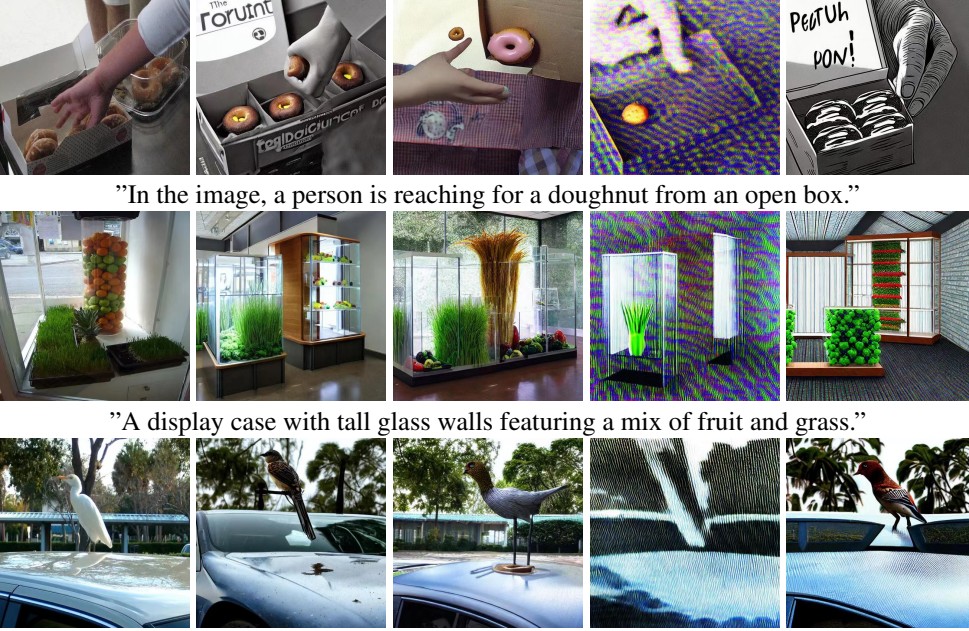

"In the image, a person is reaching for a doughnut from an open box."

"A display case with tall glass walls featuring a mix of fruit and grass."

" A car roof with a bird standing on it near a covered pavilion."

Figure 10: More qualitative results of adversarial images and their output of the SD-v1-5 image variation model.

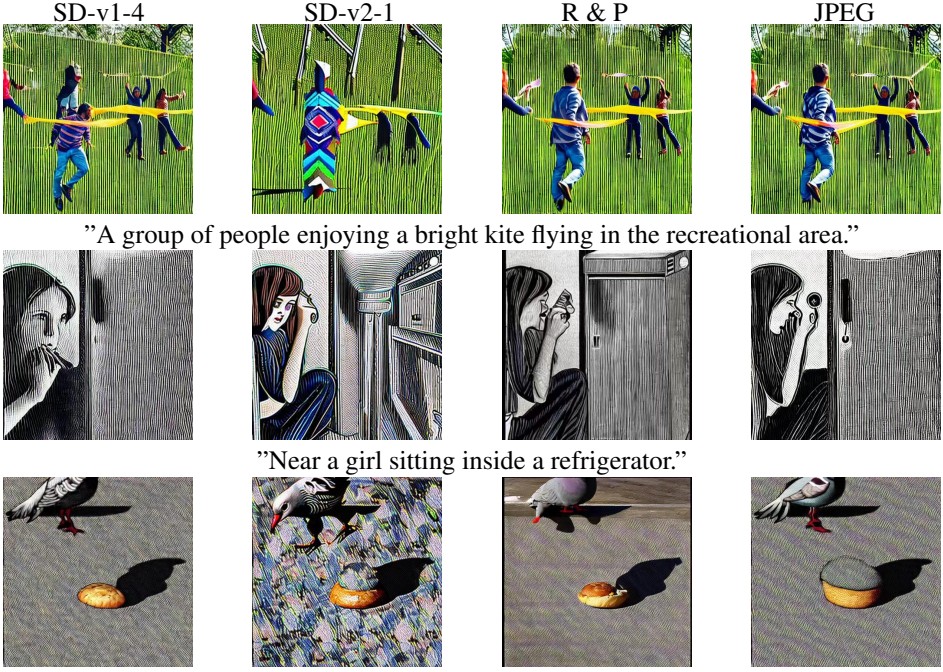

"A group of people enjoying a bright kite flying in the recreational area."

"Near a girl sitting inside a refrigerator."

"The pigeon is dining from a bread bowl, savoring every bite."

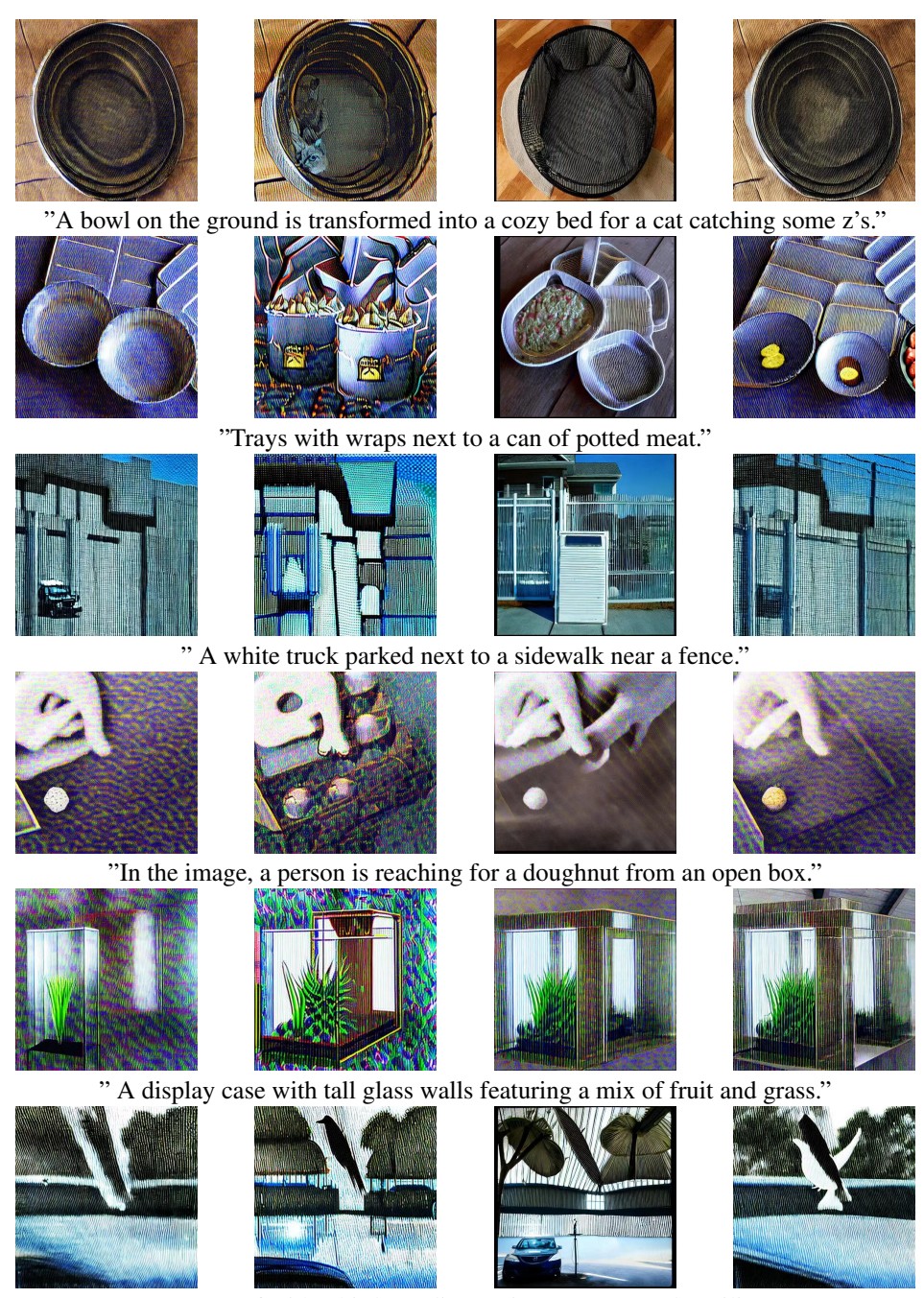

Figure 11: More qualitative results of generated images by model-transfer attacks on SD-v1-4 and SD-v2-1, and generated adversarial images of the SD-v1-5 against defense mechanisms.

## G    DISCUSSION

### G.1    USAGE

Adversarial attacks can trigger bugs inside deep neural networks, and such technologies can be misused in the real world. However, it is important to understand the inner bugs inside the diffusion models. Our study can attract the attention of researchers and motivate the research on the robustness of diffusion models, like improving the robustness of the diffusion models and designing powerful

defense mechanisms to avoid adversarial attacks. Moreover, adversarial attack strategies can be utilized in a good way to avoid intended malicious editing by the generative models for a harmonious society.

## G.2 EFFICIENCY ISSUES

The computation complexity of the adversarial attacks on latent diffusion models is high because of multiple denoising steps. However, we take measures to reduce the computation cost, including the computation time and memory. We only consider 15 denoising steps and 15 iterations for generating the adversarial examples. The computation time for generating one image is reduced to around 1 minute, and the memory is reduced under 32G. We regard that more efficient adversarial attack algorithms are required to further reduce computation complexity in the future.

## G.3 LIMITATION

The limitation of our work is that we propose to destroy all the internal features of the target module in the denoising process. However, we have no guarantee that attacking all the features is the optimal strategy. It is possible that better results are achievable by attacking part of the steps, but this would require tuning the step combination. We leave it to future work.

