# OpenReview forum: "On the Robustness of Latent Diffusion Models"
_ICLR.cc/2024/Conference — Submitted to ICLR 2024_

### Official Review · Reviewer_gYf9 · 2023-10-31

**Soundness:** 3 good
**Presentation:** 3 good
**Contribution:** 2 fair
**Rating:** 5
**Confidence:** 3

**Summary:**

This paper tries to thoroughly explore the robustness of latent diffusion models. By decomposing the latent diffusion models into different components e.g. encoder, ResNet, denoising, and decoder, we can attack each module by maximizing the changes in output. For white box settings, through experiments, the authors find that attacking the output of the resnet seems to be more effective than other methods. Also, the authors run black box attacks for the first time for LDM, using prompt-transfer attacks and model-transfer attacks.

**Strengths:**

- This paper aims to explore the robustness of LDM, which is an important problem in the age of large generative models
- The idea of decomposing the DM into sub-modules is good, by exploring each module, we can get some new insights
- The authors did extensive experiments to support their conclusions
- The paper is well-written and is easy to read

**Weaknesses:**

- For the white box settings (based on gradient):

  (1) Many attacks on LDM have been studied [1, 2], but are not mentioned in this paper.

  (2) Attacking the output of the encoder is not an optimal way, previous work tried to minimize the distance between encoded adv-samples and a target image (e.g. some noise or given image) [2]

  (3) Will the combination of the objective functions of different sub-modules be a stronger attack? This point is not discussed


 - For the black box settings: should study the transferability of unconditioned attacks without prompt


- This paper lacks many important literature reviews e.g. adversarial samples for the diffusion model [1, 2] and diffusion model for adversarial samples [3, 4], which should be mentioned and compared

[1] Adversarial Example Does Good: Preventing Painting Imitation from Diffusion Models via Adversarial Examples

[2] Glaze: Protecting artists from style mimicry by text-to-image models

[3] Diffusion-Based Adversarial Sample Generation for Improved Stealthiness and Controllability

[4] Diffusion Models for Imperceptible and Transferable Adversarial Attack

**Questions:**

- For eps=0.1, what is the range of input, [-1, 1] or [0, 1]
- How exactly is the attack conducted since the computational graph is chained with many U-Net, do we just calculate the final output of a target module? Can we attack the expectation over T of the given objective?
- In 3.4.2 why we need this:  `For the purpose of successful image editing,
we rank and select the top-5 text prompts by the CLIP score between the generated prompt and the
output image of Stable Diffusion V1-5 Rombach et al. (2022).`


I am willing to discuss and change my score if my questions in `Weaknesses` and `Questions` can be solved.

---

> ### Author Response · Authors · 2023-11-20
>
> We appreciate your valuable comments and suggestions.
>
> Q1: More references.
>
> We aim to analyze the robustness of diffusion models, and have discussed closely related works in our paper. The references you provided are not closely related to our work. [1, 2] deploy adversarial examples to protect the copyright, and [3, 4] utilize diffusion models to generate adversarial examples. We have added these references in the updated version of our paper.
>
>
> Q2: Attacking encoder.
>
> We do not claim that attacking the output of the encoder is the optimal way for attacks. Based on our experiments, we find that the Resnet in the denoising process is the most vulnerable component.
> We compare our method with Glaze [2] and DiffAttack [4] in the following table, which shows that our attacking strategy is better.
>
> | Methods | CLIP | PSNR | SSIM | MSSSIM | FID | IS |
> | :-----: | :-----: | :-----: | :-----: | :-------: | :----: | :----: |
> | Glaze | 30.32 | 12.06 |0.132 |0.344 |196.6 |17.34 |
> | DiffAttack | 32.56 | 14.52 |0.218 |0.463 |172.8 |19.53 |
> | Ours |   **29.89** | **11.82** |**0.076** |**0.270** |**206.3** | **16.93** |
>
>
>
> Q3: Combining sub-modules.
>
> We do ablation studies on combining sub-modules. We consider combining the Resnet with the encoder, FF, and decoder, respectively. As shown in the table, combining Resnet with FF improves the attacking performance, which shows the effectiveness of combining sub-modules. However, combining Resnet with the encoder or decoder decreases the attacking performance, which implies that we should carefully combine sub-modules to improve attacking performance.
>
> | Module | CLIP | PSNR | SSIM | MSSSIM | FID | IS |
> | :------: | :-----: | :-----: | :-----: | :-------: | :----: | :----: |
> | Resnet |   29.89 | 11.82 |0.076 |0.270 |206.3 |16.93 |
> | Resnet + FF |   29.62 | 11.73 |0.068 |0.252 |204.8 |16.82 |
> | Resnet + Encoder |   32.28 | 13.26 |0.132 |0.352 |189.2 |15.62 |
> | Resnet + Decoder |   32.62 | 13.87 |0.126 |0.340 |192.8 |18.11 |

---

> > ### Comment · Reviewer_gYf9 · 2023-11-20
> > **About related works**
> >
> > I know [1, 2] well, and [1, 2] aims to generate adversarial samples for diffusion models, which are highly related.
> >
> > Please correct me if I have any misunderstanding.

---

> > > ### Comment · Reviewer_gYf9 · 2023-11-20
> > > **About Attacking the Encoder**
> > >
> > > I am glad with the response of the authors to most of the questions.
> > >
> > > Attacking the encoder can actually work really well, I have some experience with it. [a, b] shows strong results based on attacking the encoder, but the right way is to drag the output of the encoder closer to a target image, and using Eq (2) in the paper is not a good way.  I will not say it is a clear flaw of this paper, but I do not want it to mislead the community, since attacking the encoder can achieve strong attacks.
> > >
> > >
> > > [a]: Mist: Towards Improved Adversarial Examples for Diffusion Models
> > > [b]: Raising the Cost of Malicious AI-Powered Image Editing

---

> ### Author Response · Authors · 2023-11-20
>
> Q4: Prompt setting.
>
> We aim to study the robustness of text-guided image editing models, which all need prompts. Therefore, it is out of the scope of this paper to explore unconditioned attacks without prompts.
>
> Instead, we consider the prompt-transfer setting to check whether the adversarial examples can mislead the model under other similar prompts (not the original one). The prompt-transfer setting represents a black-box setting, where attackers do not know the exact prompts that users adopt.
>
> As shown in Table 3 of the paper, the adversarial examples are still effective under the prompt-transfer setting, which demonstrates that the adversarial examples do not overfit to a specific prompt.
>
>
> Q6: Input range.
>
> The range of the input is [0, 1]. The qualitative examples of the generated examples are shown in Figure 3 of the paper. We can see that the adversarial perturbations are imperceptible to humans.
>
> Q7: Attack implementation.
>
> In our implementation, we attack the expectation over T of the given objective. As shown in the provided source code, the intermediate results of the chained U-Net are stored for the computation of the gradient.
>
>
> Q8: Section 3.4.2.
>
> Before we launch adversarial attacks, we should ensure that the model can correctly edit the image based on the guidance of the text prompt. Otherwise, the evaluation of adversarial attacks would be inaccurate, because we cannot tell whether the malfunction of the model is due to an adversarial example or the original input. Therefore, we do this step to ensure that models function correctly on the original inputs before we use them to generate adversarial examples.
>
>
> [1] Liang C, Wu X, Hua Y, et al. Adversarial Example Does Good: Preventing Painting Imitation from Diffusion Models via Adversarial Examples, ICML 2023.
>
> [2] Shan S, Cryan J, Wenger E, et al. Glaze: Protecting artists from style mimicry by text-to-image models, USENIX Security 2023.
>
> [3] Xue H, Araujo A, Hu B, et al. Diffusion-Based Adversarial Sample Generation for Improved Stealthiness and Controllability, Arxiv 2023.
>
> [4] Chen J, Chen H, Chen K, et al. Diffusion Models for Imperceptible and Transferable Adversarial Attack, Arxiv 2023.

---

> ### Author Response · Authors · 2023-11-22
> **About related works**
>
> We have already included [1, 2] in the updated version of the paper.
>
> The goal of [1,2] is different from our work. [1,2] aim to protect the copyright. To this end, [1,2] endeavor to find successful adversarial samples against diffusion models. In contrast, we aim to analyze the robustness of diffusion models. To this end, we analyze the robustness of modules inside the diffusion model and also consider different transfer settings.

---

> > ### Comment · Reviewer_gYf9 · 2023-11-22
> > **Thanks for your response.**
> >
> > I do not quite get it, since this paper aims to generate adv-samples to fool the LDM (use some losses depending on different modules), and [1, 2] also aims to generate adv-samples to fool the LDM.
> >
> > [a] works much better than [b] by choosing a proper target, and shows to be strong in many settings. I do not expect the experiments on [a], but I think it is necessary to clarify why [a, b] are not included as a baseline or something, since basically they are all doing adv-attacks to a LDM, which can reveal the robustness of a LDM.

---

> > > ### Author Response · Authors · 2023-11-23
> > >
> > > Thank you for your reply. We have added the references [1,2,a,b] in the updated version of our paper. If it is necessary, we will include the comparison experiments in our paper.

---

> ### Author Response · Authors · 2023-11-22
> **About Attacking the Encoder**
>
> We do not claim that attacking the encoder fails to work well. We just claim that attacking the encoder is not the best way for attacks. As shown in Table 1 of the reference [b], attacking the whole diffusion process (i.e., the decoder) is better than attacking the encoder. Therefore, [b] has already demonstrated that attacking the encoder is not the best way for attacks.
>
> Additionally, we do experiments to compare our method with the Encoder attack in [b]. The results show that attacking the encoder is inferior than attacking the Resnet module inside the denoising steps.
>
>
> | Methods | CLIP | PSNR | SSIM | MSSSIM | FID | IS |
> | :-----: | :-----: | :-----: | :-----: | :-------: | :----: | :----: |
> | Encoder [b]| 31.14 | 12.90 |0.153 |0.316 |182.1 |16.99 |
> | Resnet (Ours) |   **29.89** | **11.82** |**0.076** |**0.270** |**206.3** | **16.93** |
>
> We think that our method that uses Eq (2) to attack is reasonable, since the core idea of our method and attacking the encoder is similar. Both [a,b] and our approach aim to disrupt the current feature representation, which can undermine the model’s output in the end.
> Specifically, [a,b] drag the output of the encoder closer to a target image to destroy the feature representation, while we push apart the representations of the adversarial example and the original example to destroy the feature representation.

---

> ### Author Response · Authors · 2023-11-22
>
> Dear Reviewer,
>
> Considering that the discussion phase is nearing to end, we are looking forward to your further feedback about our latest response. Do our responses fully address your concerns? Do you have any other comments? We would like to discuss with you in more detail. We greatly appreciate your time and feedback.
>
> Sincerely,
>
> Authors

---

### Official Review · Reviewer_ibVi · 2023-10-31

**Soundness:** 2 fair
**Presentation:** 3 good
**Contribution:** 2 fair
**Rating:** 5
**Confidence:** 5

**Summary:**

The paper presents a comprehensive study on the robustness of Latent Diffusion Models (LDMs), specifically focusing on their vulnerability to adversarial attacks. The authors delve into the components of LDMs, pinpointing the denoising process, and more precisely the ResNet architecture, as the most susceptible to these attacks. In addition to this vulnerability assessment, the paper makes significant strides in dataset construction, offering an automated pipeline that facilitates the evaluation of LDMs under adversarial conditions. However, to enhance its academic rigor and impact, the authors should address the highlighted weaknesses and consider the posed questions for future work. I recommend this paper for acceptance, contingent upon the incorporation of these suggested improvements.

**Strengths:**

1. Robustness Evaluation: The paper excels in providing a thorough investigation of the robustness of LDMs, addressing both white-box and black-box adversarial attacks. This dual perspective enriches the paper's contributions and sets a solid foundation for future research in this domain.

2. Dataset Construction: The introduction of an automatic dataset construction pipeline is a noteworthy contribution, as it streamlines the process of evaluating LDMs and ensures a consistent and reproducible framework for future studies.

3. Comprehensive Attack Analysis: Including black-box adversarial attacks alongside the white-box attacks offers a more holistic view of the vulnerabilities in LDMs, showcasing the paper's commitment to a comprehensive analysis.

4. Open-Sourced Results: The decision to open-source the results demonstrates transparency and fosters a collaborative environment, enabling other researchers to build upon this work.

**Weaknesses:**

1. Methodological Clarity: The paper could benefit from a more explicit elucidation of the adversarial attack strategies employed. Diving deeper into the rationale behind each attack, the expected impacts, and the choice of specific models would significantly enhance the reader's understanding and the paper's overall impact.

2. Limited Scope: Focusing solely on LDMs narrows the breadth of the paper's contributions. Expanding the analysis to include comparisons with other models could provide a more comprehensive understanding of the robustness landscape.

3. Novelty Concerns: While the dataset construction is a notable effort, the evaluation of LDMs in this context is somewhat straightforward. A more innovative approach or unique insights into the robustness of LDMs would elevate the paper's significance.

4. Presentation and Formatting: The paper could improve in terms of presentation, including clearer figures, more concise explanations, and better-structured arguments to enhance readability and comprehension.

**Questions:**

1. Resource Utilization: Could you provide more details on the GPU consumption and running time associated with your method? Understanding the computational efficiency of your approach is crucial for practical applications.

2. Defensive Strategies: Have you explored or considered any novel defense mechanisms to bolster the robustness of LDMs against adversarial attacks? While this might extend beyond the current scope of your work, including discussions or suggestions in this area could significantly strengthen your paper.

---

> ### Author Response · Authors · 2023-11-20
>
> We appreciate your valuable comments and suggestions.
>
> Q1: Methodology.
>
> In this paper, we launch adversarial attacks on representative modules inside the latent diffusion models to analyze their adversarial vulnerability, which is under-explored and important for understanding the robustness of diffusion models. Therefore, we delve into the three stages of the latent diffusion models: encoding, denoising, and decoding, which are illustrated in Figure 1 of the paper. In order to comprehensively analyze the robustness of each module inside the network, we propose to destroy the latent features of each component as shown in Equation 2 of the paper. We first study the white-box attack performance. We find that the Resnet in the denoising process is the most vulnerable component of the latent diffusion model. Then, we further analyze the transfer attacks and possible defenses. We find that adversarial examples are still effective under the model-transfer and prompt-transfer settings, and against possible defenses, which raises a concern about the robustness of diffusion models.
>
> Q2: Other models.
>
> Please see Q1 of Reviewer Lyr6.
>
>
> Q3: Novelty.
>
> We construct a benchmark dataset to facilitate the fair comparison of different attacks for future research.
> Besides, we are the first to comprehensively analyze the adversarial robustness of each module inside the latent diffusion models, while previous approaches fail to explore the denoising process and the internal modules of the latent diffusion models.
> Furthermore, we are the first to explore the transfer attacks against latent diffusion models, including the model-transfer and prompt-transfer settings. We also explore possible defenses for diffusion models. We have several new findings based on our exploration. For example,
>
> (1)	The Resnet in the denoising process is the most vulnerable component of the latent diffusion model.
>
> (2)	Adversarial examples are effective across models and similar prompts.
>
> (3)	SD-v2 is more vulnerable than SD-v1, and the vulnerabilities of SD-v1 are inherited by SD-v2.
>
> (4)	Adversarial attacks are still effective against defense methods.
>
> We believe that all these findings can facilitate future research on the robustness of latent diffusion models.
>
> Q4: Presentation.
>
> We have improved the presentation in the updated version of our paper.
>
> Q5: Complexity.
>
> Under our experimental setting, we generate one adversarial example in about 47.6 seconds, and the time complexity can be further reduced by decreasing the generating iterations and reducing the denoising steps. In addition, the generated adversarial examples are transferable across models and prompts as we discussed in the paper, so they can be directly used to test other diffusion models after one-off generation. The memory used is about 29.6 GB, which is also available on a GeForce RTX 3090 GPU. The memory costs can also be reduced by reducing the denoising steps.
>
>
> Q6: Possible novel defenses.
>
> Based on our finding that the Resnet in the denoising process is the most vulnerable component of the latent diffusion model. We can build possible novel defenses from a structure perspective. For example,
>
> (1)	We can replace the vulnerable components to improve the robustness of LDMs. [1, 2] discard the Resnet component in the transformer, showing better robustness compared with SD.
>
> (2)	We can also add additional filter layers before the vulnerable components, such as Resnet and FF, to reduce the adversarial noise.
>
>
> [1] Bao F, Nie S, Xue K, et al. One transformer fits all distributions in multi-modal diffusion at scale, ICML 2023.
>
> [2] Xu X, Wang Z, Zhang G, et al. Versatile diffusion: Text, images and variations all in one diffusion model, ICCV 2023.

---

> ### Author Response · Authors · 2023-11-22
>
> Dear Reviewer,
>
> Considering that the discussion phase is nearing to end, we are looking forward to your further feedback about our latest response. Do our responses fully address your concerns? Do you have any other comments? We would like to discuss with you in more detail. We greatly appreciate your time and feedback.
>
> Sincerely,
>
> Authors

---

### Official Review · Reviewer_Lyr6 · 2023-10-31

**Soundness:** 3 good
**Presentation:** 3 good
**Contribution:** 3 good
**Rating:** 6
**Confidence:** 3

**Summary:**

This paper explores the robustness of diffusion models from the perspective of adversarial attacks. Firstly, the authors compare the white-box robustness of latent diffusion models used for image editing, demonstrating the most vulnerable components within these diffusion models. Furthermore, they consider two transfer-based black-box scenarios. Finally, they propose an automated data set construction pipeline for building a high-quality, publicly available dataset.

**Strengths:**

1.The paper addresses the robustness of latent diffusion models, which is highly significant.

2.The paper explores the structure of diffusion models beyond attacking the encoder and discusses black-box attack settings.

3.The proposed attack pipeline is simple and easy to follow.

**Weaknesses:**

1.	Lack of comparison with other types of diffusion models。The experiments in the paper primarily focus on different versions of Stable Diffusion, lacking comparisons and discussions regarding other types of diffusion models. This may not comprehensively represent all latent diffusion models. I suggest the authors consider comparing with other models, such as UniDiffuser[1].

2.	Limited exploration of diverse attacks and defenses。As the paper aims to explore the robustness of diffusion models from the perspective of adversarial attacks, the paper uses a limited range of attack methods, and the design of attacking method may lack novelty. Additionally, there is no new attack strategy developed based on the unique characteristics of diffusion. Furthermore, the defense methods chosen for experiments appear outdated. I recommend the authors consider incorporating a wider variety of attacks and defenses in their experiments.

3.	Insufficient discussion of denoising steps. In the experiments, the authors only consider 15 denoising steps for generating adversarial examples. However, according to the findings in DiffPure[2], longer denoising steps introduce randomness and enhance robustness. The authors could explore the impact of different denoising steps as a hyperparameter on diffusion model robustness.

[1]Bao F, Nie S, Xue K, et al. One transformer fits all distributions in multi-modal diffusion at scale[J]. arXiv preprint arXiv:2303.06555, 2023.
[2]Nie W, Guo B, Huang Y, et al. Diffusion Models for Adversarial Purification[C]//International Conference on Machine Learning. PMLR, 2022: 16805-16827.

**Questions:**

see the weakness

---

> ### Author Response · Authors · 2023-11-20
>
> We appreciate your valuable comments and suggestions.
>
> Q1: Other models.
>
> We experiment with two other types of diffusion models, including Unidiffuser [1] and Versatile Diffusion [2]. We follow the experimental setting of Table 2 in the paper. The results are as follows. From the table, we can draw a similar conclusion that other types of diffusion models are also susceptible to adversarial attacks, and the denoising process is the most vulnerable part.
>
> | Model | Module | CLIP | PSNR | SSIM | MSSSIM | FID | IS |
> | :------: | :------: | :-----: | :-----: | :-----: | :-------: | :----: | :----: |
> |                      | Encoding |   33.74 | 13.14 |0.166 |0.394 |192.8 |17.63 |
> | Unidiffuser  | Denoising |   **31.82** | **11.53** |**0.102** |**0.297** |**212.3** |**13.93** |
> |                      | Decoding |   33.62 | 12.92 |0.154 |0.379 |201.2 |16.12 |
> |                      | Benign |   34.38 | $\infty$ |1.000 |1.000 |174.1 |21.92 |
> |                      | Encoding |   33.91 | 13.61 |0.186 |0.407 |179.3 |18.63 |
> | Versatile  | Denoising |   **32.82** | **11.73** |**0.132** |**0.355** |**197.1** |**15.11** |
> |                      | Decoding |   34.03 | 13.22 |0.172 |0.386 |183.3 |17.82 |
> |                      | Benign |  34.66 | $\infty$ |1.000 |1.000 |163.6 |22.61 |
>
> Q2: More attacks and defenses.
>
> We compare our approach with more attack methods, including FDA [3], NAA [4], Glaze [5] and DiffAttack [6]. From the experimental results, we can see that our approach also outperforms the advanced attack methods, which validates the superiority of the employed attack method.
>
> | Methods | CLIP | PSNR | SSIM | MSSSIM | FID | IS |
> | :-----: | :-----: | :-----: | :-----: | :-------: | :----: | :----: |
> | FDA | 32.18 | 13.28 |0.209 |0.442| 176.8|18.32 |
> | NAA | 31.46 | 13.01 |0.172 |0.390 |185.8 |17.47 |
> | Glaze | 30.32 | 12.06 |0.132 |0.344 |196.6 |17.34 |
> | DiffAttack | 32.56 | 14.52 |0.218 |0.463 |172.8 |19.53 |
> | Ours |   **29.89** | **11.82** |**0.076** |**0.270** |**206.3** | **16.93** |
>
> We also experiment with two defenses: EBM [7] and DiffPure [8]. From the experimental results, we can see that our attacks are still effective against advanced defenses.
>
> | Methods | CLIP | PSNR | SSIM | MSSSIM | FID | IS |
> | :-----: | :-----: | :-----: | :-----: | :-------: | :----: | :----: |
> | EBM  | 33.62 | 12.02 |0.160 |0.386| 180.2|21.62 |
> | DiffPure  | 34.15 | 12.54 |0.209 |0.433 |184.1 |23.17 |
> | Benign  | 34.74 | $\infty$ |1.000 |1.000 |167.9 |19.86 |

---

> ### Author Response · Authors · 2023-11-20
>
> Q3: Denoising steps.
>
> We do an ablation study to explore the impact of different denoising steps (5, 10, 15, 20, and 25) on the robustness of diffusion models. As we can see from the table, the performance increases with the increase of denoising steps, but the computational complexity also increases dramatically. Therefore, we select the denoising step to be 15 in our experiment to balance the performance and the computational cost.
>
> | Steps | CLIP | PSNR | SSIM | MSSSIM | FID | IS |
> | :-----: | :-----: | :-----: | :-----: | :-------: | :----: | :----: |
> | 5   |  33.85 | 14.40 |0.240 |0.436 |180.91 |17.78 |
> | 10 |   31.86 | 12.86 |0.164 |0.344 |195.42 |17.01 |
> | 15 |   29.89 | 11.82 |0.076 |0.270 |206.3 |16.93 |
> | 20 |   29.86 | 11.76 |0.076 |0.268 |206.5 |16.85 |
> | 25 |   29.76 | 11.62 |0.074 |0.261 |207.2 | 16.75 |
>
>
>
> [1] Bao F, Nie S, Xue K, et al. One transformer fits all distributions in multi-modal diffusion at scale, ICML 2023.
>
> [2] Xu X, Wang Z, Zhang G, et al. Versatile diffusion: Text, images and variations all in one diffusion model, ICCV 2023.
>
> [3] Ganeshan A, BS V, Babu R V. Fda: Feature disruptive attack, ICCV 2019.
>
> [4] Zhang J, Wu W, Huang J, et al. Improving adversarial transferability via neuron attribution-based attacks, CVPR 2022.
>
> [5] Shan S, Cryan J, Wenger E, et al. Glaze: Protecting artists from style mimicry by text-to-image models, USENIX Security 2023.
>
> [6] Chen J, Chen H, Chen K, et al. Diffusion Models for Imperceptible and Transferable Adversarial Attack, Arxiv 2023.
>
> [7] Hill M, Mitchell J, Zhu S C. Stochastic security: Adversarial defense using long-run dynamics of energy-based models, ICLR 2021.
>
> [8] Nie W, Guo B, Huang Y, et al. Diffusion models for adversarial purification, ICML 2022.

---

> ### Author Response · Authors · 2023-11-22
>
> Dear Reviewer,
>
> Considering that the discussion phase is nearing to end, we are looking forward to your further feedback about our latest response. Do our responses fully address your concerns? Do you have any other comments? We would like to discuss with you in more detail. We greatly appreciate your time and feedback.
>
> Sincerely,
>
> Authors

---

### Meta-Review · Area_Chair_17Gw · 2023-12-08

**Metareview:**

The paper studies the robustness of diffusion models towards adversarial attacks. The paper is well presented and easy to follow. The reviewers therefore gave scores close the a borderline rating with a tendency for rejection. After the rebuttal and discussion phase, there are  still remaining concerns regarding the delineation to prior work. Besides, a more careful discussion of the potential conclusions regarding attacks on the encoder of diffusion models will be beneficial. With these aspects in mind, the submission is not ready for acceptance at ICLR.

**Justification For Why Not Higher Score:**

The paper received borderline scores with a tendency to rejection. The paper is not the first to analyze the robustness of latent diffusion models and fails to clearly delineate itself from prior methods. While other aspects such as the clarity in presentation can be easily fixed, the paper will need a major revision to be ready for acceptance.

**Justification For Why Not Lower Score:**

N/A

---

### Decision · Program_Chairs · 2024-01-16

Reject